# Exploring the Alignment between Digital Strategies and Educational Practices in Higher Education Infrastructures

Egil Øvrelid

Department of Informatics, University of Oslo, 0373 Oslo, Norway; egilov@ifi.uio.no

**Abstract:** Higher education is a key pillar in constructing new knowledge economies for the 21st century, and the digitalization of higher education is a central focus area for national authorities. Visionary discourses from authorities state that the decision-making authority for digital strategies should be centralized to the domain of management. Digitalization is, however, driven by key features of modern technology and may also lead to the transformation of traditional educational methods as well as educational practices. Since the university contains several disciplines, different strategies can be used when products or processes within the disciplines are digitalized. It is important to consider in the ways that different disciplines can proceed to digitalize their educational practices. Based on these interests, our research question is as follows: how do digital strategies in higher education emerge, and how do they align with the educational context? Through a qualitative case study with interviews, participation in workshops, and document analyses, we investigated two digitalization efforts in the fields of medicine and law. We found that the two classical disciplines' strategic approaches differed substantially. Based on the findings, our main contribution is a digitalization model with two archetypes, namely digital transformation strategy and digital innovation strategy. The model highlight the main object of the respective strategies, but also the institutional reaction to digitalization efforts. An implication from our study is the demonstration of how specific faculties adapt digital strategies to educational practices. This may sometimes lead to the transformation of educational practices, while other times more incremental moderate changes may be implemented. From a practical point of view, policymakers, politicians, educational management, and professionals need knowledge and expertise about the implications of digital strategies for educational practices. Our contribution, we propose, strengthens the understanding of strategies within digital infrastructures in higher education.

**Keywords:** digital strategies; higher education; educational practices; digital infrastructure



## 1. Introduction

Digitalization implies combining physical and digital artifacts to simplify and improve processes or products [1,2]. The consequences of this reconfiguration will vary. In some cases, entire sectors or organizations can be transformed [3]. Based on Wessel et al. [4] we see digital transformation as the process of transforming deeply anchored practices or objects leading to a (to some extent) new identity.

In other cases, the change will be less dramatic [5]. We refer to this as digital innovation, and define it as "the carrying out of new combinations of digital and physical components to produce novel products" [2]. Digital innovation may produce new products or processes, but more incrementally.

The source of the requirements for change can also vary; sometimes the requirements come from the outside [6,7], and other times they may emerge as requirements from the inside of the organization [8]. The literature on digitalization has demonstrated how physical equipment in cars can be digitalized [9], how services may be improved [10], as well as how business models can be transformed [11], but also the organizational cost of these activities [6,9]. The properties of digital technology make digitalization a

strategic activity, but the IS stream of the literature has to a much lesser extent focused on digitalization of higher education, where organizational issues may be different than in the competitive industry.

Higher education is a key pillar in constructing new knowledge economies for the 21st century [12]. This is why digitalization of higher education is a central focus area for national authorities [13]. As digitalization of higher education poses new challenges to the sector, a strategic viewpoint is essential [14–16]. In Norway, the strategy of the authorities holds the premise that the decision-making authority for digitalization should be at a strategic level and be integrated into all professional and administrative activities [13]. By digital strategy, we mean the activities concerning 'creation and appropriation of value, by exploiting digital technologies to achieve long-term objectives' [17].

Since the activity of digitalization has transformative potential, traditional educational methods and practices may change [18]. Bearing in mind that the university historically has an "unlimited aggregation of specialties" (informatics, economics, medicine, law, various social sciences, language courses, pedagogy, psychology, and so on) [19] (p. 14), and that each discipline has autonomous control regarding its organization, different strategies can be used to digitalize the various disciplines [20]. For instance, each particular discipline may prioritize maintaining educational standards, and the high-level strategies from the authorities may, thus, experience counter-strategies [15]. A better understanding of the chosen strategies is needed to understand how national digitalization initiatives are balanced towards the independent assessments and decisions of each discipline. Since strategic approaches to digitalization may differ in each discipline we ask the following question: how do digital strategies in higher education emerge, and how do they align with the educational context?

We investigated two classical disciplines, namely law and medicine, at the biggest university in Norway. Both have a historical tradition of employing the entrepreneurial mindset seen as an important fundament for digitalization of higher education [15]. To develop our argument, we frame the research within digital infrastructure theory [21,22], both to identify contextual challenges that condition and form the strategic approach, and to identify how these challenges change or transform the digital infrastructure. Our main contribution is two models that contribute to understanding, explaining, and managing strategic challenges.

## 2. Theoretical Framework

### 2.1. Digital Infrastructures

A fundamental understanding of information systems requires taking into account both the technology, the organization, and the individual agency, and their collective action in dealing with the various requirements [23]. In addition, the increasingly networked information and communication channels of modern organizations make it advantageous to see information systems as digital infrastructures [21,22], where systems, organizations, and agencies are interconnected in a way that makes it important to see their contextual contingencies.

A central interest within the infrastructure literature has been to understand the patterns and mechanisms by which infrastructures evolve [21]. While some instances of the literature frame the evolution as a managed alignment between IT capabilities and the business processes [24], others are rather occupied with the radical emergence of evolutionary paths through serendipitous innovation outside management control [25]. Work inspired by this last tradition often focuses on evolution as something that is formed inside the organization in the different practice environments that operate there and the role they have in shaping the evolution [8,26]. A third and relatively recent stream of research within this literature is occupied with how strategic planning and the digital infrastructure are aligned to enable innovation, adoption, and scaling [6,21].

*2.2. Addressing Strategic Challenges in Digital Infrastructure Evolution*

The digital infrastructure literature highlight three key strategic challenges in IS implementation, namely to define a competitive strategy and achieve alignment between strategy and IT capabilities, to align the strategy with the related practices to obtain actual alignment and to ensure that strategy is a sustained activity [5]. This has several implications for digital infrastructure strategy.

First, there must be a clear strategy that takes into account the opportunities afforded by the IT portfolio. The "planning literature" within IS [27,28] is occupied with how management identifies new trends and tendencies. Recently a similar business oriented stream of the literature has claimed that strategies must be planned, initiated, managed, and maintained by some form of central leadership [3,29]. The management will also have to take into account the digital resources, which are the existing IT components, architectures, and products [30], when defining and implementing strategic plans.

Secondly, an organization is adapted to different purposes and has different properties to solve these purposes. This implies that strategy must take into account the internal organization of the existing firm and the various knowledge workers' epistemic practices [31]. Moreover, in fields where specialized professional knowledge dominates, the decentralized autonomy is notable, as well as a low degree of transparency and a high degree of uncertainty [32]. If the internal knowledge workers or activity systems [33] have a rather autonomous culture, the strategies chosen should be aligned accordingly [34].

This means that the strategy must take the knowledge workers and activity systems into account in the digitalization activity, but since the practices have a different impact on digital strategy, the degree of inclusion and participation in the change process will vary [35]. However, it is a management challenge to ensure that different stakeholders contribute to the change process by moving "forward using their differences, in a productive rather than in a fractious way" [36] (p. 283).

Third, the strategy must be a sustained activity, where the governance of the infrastructure at the administrative level secures continuity and controls interruptions [37]. Management has a particular responsibility, since emergent requirements, such as new educational standards or new research areas, need to be taken care of continually [38].

Our goal in this paper is to understand strategies for the digitalization of digital infrastructures in higher education, and the contextual challenges that form the strategy. To gain knowledge of how IS strategies are formed to solve particular challenges in higher education, we investigated two classical disciplines, namely law and medicine. We will describe our findings after the method chapter, which follows.

## 3. Method

Our case study was performed at the largest university in Norway, with more than 27,000 students and 6000 employees. The university has recently initiated a digitalization strategy that implies more centralized governance of core IT systems. We aimed to investigate how digital strategies in higher education emerge, and to what extent they are aligned with educational practices, in order to conceptually distinguish between digital transformation strategies and more incremental approaches. We investigated several empirical cases from humanities, pedagogics, informatics, law, and medicine. Comparing the cases, we found that the two classical disciplines of law and medicine had several similarities concerning their approaches and goals, but clear differences regarding the speed of implementation and alignment with educational practices (see Table 1). The law case concerned the transition from manual books to digital sources in education. The case from medicine concerned the implementation and use of an e-learning system. Both faculties were hailed for their innovative work and, through our access, we were able to observe the materialization of this work in action. In addition, the two cases experienced substantial pressure from internal and external stakeholders to speed up their digitalization initiatives, and we were curious to identify the main consequences of this pressure on the

digital strategies. In relation to speed, the two cases expose alternative strategic routes, and we were able to build on these alternatives to deduce general strategies for each of them.

**Table 1.** Overview of cases (2020).

| Case | Trigger | Aim | Result |
|---|---|---|---|
| E-learning in medicine | Teachers want to improve students' mastery of communication, practical procedures, visual analysis, and clinical decision-making. Students want more digital feedback. | To facilitate a better and faster learning process to educate better doctors | The number of users is not known |
| Digital sources of law | The market requires more digital competency amongst students of law. | To educate law students with more digital competency | 4500 users by the end of 2019 |

We chose a longitudinal process study [39] to study the phenomena over time and to investigate the longitudinal interaction between organizations, humans, and technology. Our case study research approach is based on engaged scholarship [40] where informants are not only sources of empirical data, but also helpful in constructing narratives and discussing theoretical and practical implications.

### 3.1. Data Collection

The data collection was carried out between January 2019 and June 2020 but included a reconstruction of historical data. Thirteen interviews were conducted in addition to document and web-page analyses. Medical doctors, deans, teachers, students, and administrators were interviewed. Moreover, the researcher gained access to student groups and observed their interactions and debates for 3 h.

A range of documents and digital resources, such as web pages, short films, and slides were analyzed. The documentation includes policies, project manuals, and progress plans. Finally, several discussions regarding findings and possible interpretations were performed in seminars and workshops. Moreover, the researcher was part of a research group on higher education where the findings were discussed and feedback was given.

### 3.2. Data Analysis

In the analysis (see Table 2), we first established a chronology of important events. Building on Langley's (1999) approach on process data, we analyzed the historical background of the projects and were especially interested in the technology initiatives and how they were related to specific key events. The analysis revealed three central activities. First, we inspected the planning activity and identified the existing digital resources. Then, we focused on the various activities performed to align the plan with the professional workers and the students. Lastly, we were able to see how the strategy took into consideration systematic follow-up and maintenance of the digital infrastructure.

**Table 2.** Data analysis.

| Step | Description | Output |
|---|---|---|
| 1 | Identify key events, key objects in the history of the faculty | Timeline for each project |
| 2 | Analyze cases of planning and digital resources, align strategy and professional workers, follow up and maintain | Section 5 |
| 3 | Propose two models to describe the core content of strategies | Section 6 |

We see both cases as performed within digital infrastructures, since a huge amount of students, teachers, administrators, and developers, as well as resources, routines, processes, and activities from a range of stakeholders, are involved. In the case of law, more than

4500 students use the system regularly, while in medicine over 1000 students are using the e-learning infrastructure.

Lastly, based on our findings, we create and discuss two different strategic models. While the key strategic activity in our analyses (see Table 3) is derived from our framework, the models are based on elements from our empirical data but theorized in accordance with our framework. We first identified five key topics. The topics were identified through an abduction process where we suggested and tested the key object of the strategies, and the central strategy activities (key, type, and driver, see Table 4). Then we observed that the institutional reaction to the implementation was significant but very different in the two cases. This enabled us to reflect on the institutional consequences of the respective models.

**Table 3.** Comparison between digitalization initiatives in medicine and law.

| Strategic Activity | Element | Digital Infrastructure E-Learning | Digital Infrastructure Sources of Law |
|---|---|---|---|
| Planning | Trigger and driver | Professional culture | The professional lawyer. Market. IT |
| | Role of the Faculty | Supportive regarding strategy, governing regarding technology | Governing both strategy and technology |
| | Digital resources | Visualization and sound. Web pages | Text. Sources of law. Lovdata |
| Alignment | Alignment between plan and professional work | High inclusion. Each medical area chooses whether to develop and use e-learning. Adaptation: trial and error. | Low/general inclusion. Centralized development and implementation. Adaptation through training. |
| Maintenance and Follow-up | Governance | Section for medical informatics | Lovdata, an external foundation, and the management at the Faculty of law |

**Table 4.** Comparison of the two models.

| Topics | Digital Transformation Strategy | Digital Innovation Strategy |
|---|---|---|
| Key object | Commercialization | Professionalization |
| Key strategic challenge | Adapt to commercial trends and requirements | Adapt to new treatment forms and educational forms |
| Type of strategic change | Fast. Top-Down | Slow. Bottom-up |
| Driver for strategic change | External requirements | Internal |
| Institutional reaction | Turbulence | Restlessness |

## 4. Findings and Analyses

The two investigated cases have a rich and interesting history with the continual use of technology to solve core challenges within the respective areas. Since the two disciplines' inner workings make a significant impact on the strategy, the various strategies may deviate from the ones espoused by the ministries. In Sections 4.1 and 4.2, we will describe the digitalization initiatives in medicine and law, respectively.

### 4.1. Digital Education in Medicine

Digitalization of the education at the Faculty of Medicine spans a wide range of activities, from developing and implementing a student information system to digital exams, support systems for lecturing and group work, and a large portfolio of e-learning programs mainly for self-study. This analysis will focus on e-learning programs.

The e-learning activity (see Figure 1) started in 1992 when two teachers developed a series of text and image-based programs to teach students basic medical procedures, such as taking a blood sample. In 2000, the Dean of Studies funded and launched a new

initiative to employ IT in education, focusing on establishing and studying net-based collaboration during clinical placement. A year later, e-learning was included as a target area. The professor of medical informatics was put in charge of IT in education. This is now a full-time activity. Two technical positions were funded, as well as an annual project call targeted at the teachers. In 2010, the section of medical informatics was established directly under the Dean to emphasize the over-arching responsibility of the team. The e-learning programs are openly published in a national infrastructure loosely shared by the Norwegian medical faculties. Today, the digital resources comprise approximately 50 smaller or larger programs that include several hundred videos and links to server-based functionalities, such as formative tests and virtual microscopy. The digital infrastructure (http://meddev. uio.no/elaringsportalen/, accessed on 10 January 2020) is currently housing an extended amount of resources from 52 disciplines. The development of new programs can best be described as an educated development in collaboration with educational scientists and the faculty's newly appointed psychometrician.

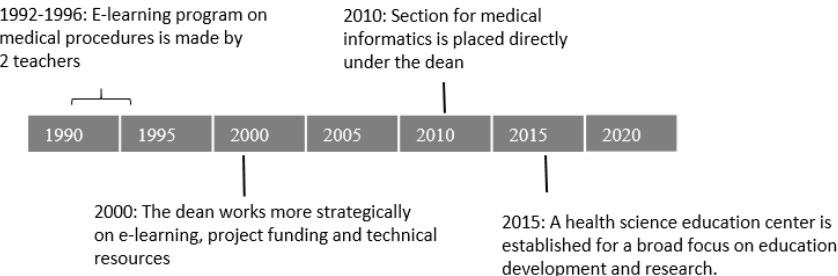

**Figure 1.** History of e-learning in the Faculty of Medicine (1992–2020).

### 4.1.1. Strategic Planning of E-Learning

The ultimate goal of e-learning and teaching as a whole is to produce highly qualified doctors. Many if not the majority of teachers are constantly looking for ways to improve the learning process and the learning outcome. Medicine is a "handicraft" full of human interaction and sensory input—visual, auditive, tactile—and complex and intertwined life processes. Learning this through books can be quite hard, whereas IT, through its ability to handle multimedia, dynamics, information networks, and interactivity, facilitates this type of learning. Students are constantly complaining about the lack of feedback on their learning progress. This is partly due to limited resources, and automated formative tests with extended feedback on the answers are seen as a partial solution to this problem. Today's students are "computer savvy" from early childhood and accustomed to IT as a natural part of their learning toolbox. An educational institution that does not employ these tools is seen as thwarting their learning. Finally, studying is becoming a more and more distributed and asynchronous activity, enforced by e.g., the advent of "flipped classrooms". Furthermore, IT is effective in distributing material in space and time. Students are expecting digital resources that facilitate individual self-studies, but also enable stronger interaction. Examples are tests, quizzes, etc., but also other forms of resources that activate the learning ability of the student.

The section for medical informatics is appointed by the faculty to take care of these issues and does this partly by funding and participating in teacher-initiated projects to develop e-learning programs. Approximately USD 150,000 is annually allotted to these projects.

> "The initiative does not come from the departments, but from the ground floor: the teachers. We try to involve students in all projects–their view is important because the product is for them, but students are usually far more than 'viewers' they often produce most of the resources under the guidance of teachers."

Even though the e-learning initiative is very popular, the resources are limited, and the making of e-learning is demanding for each discipline. Each discipline must, together

with the three employees in the section for medical informatics, do all the project work in addition to their regular work. The funding, therefore, is used almost entirely as part-time employment by students or freelance resources. The strategy is based on "cultivation" as it depends on independent activity from each subject area. This is challenging in that good initiatives rely on the discipline itself and is less anchored in faculty management.

### 4.1.2. Digital Resources

Medical education consists of 8 learning modules, and there are about 50 different e-learning programs of varying sizes and sophistication available, covering parts of all modules. The strategy from the Faculty of Medicine is two-fold. First, it is to digitalize where IT has special advantages. This applies to images, such as X-rays, eye diseases and skin diseases, and sound, for example, in auscultation training. Furthermore, movies are used for case histories, e.g., in psychiatry and clinical communication, and in procedure visualization. Animation can be used to visualize process dynamics such, as physiology and disease processes, and simulation helps to understand the processes and consequences of interventions. Thus, both practical and cognitive skills are developed. Furthermore, technology is used to "link together material in learning hierarchies so that one can go seamlessly from overview learning to in-depth learning".

Through e-learning, fragmented disciplines can obtain virtual homes that bind the fragments together in an integrated presentation. Furthermore, e-learning can also be a tool for "faculty development" where, e.g., teaching consistency is developed from a common knowledge base of procedures. The e-learning method is also used for student-activated teaching through the use of virtual patients and interactive quizzes. These many facets make e-learning an integrated knowledge system. In 2018, approximately 77,000 quizzes and virtual patient cases were delivered. In addition, approximately 170,000 slides were made available.

Figure 2 is an example that demonstrates how to place the probe and to inspect the resulting ultrasound pictures. The e-learning programs, such as those demonstrated in Figure 2, are developed and tested through trial and error. The section for medical informatics is trying to build educational tools based on what is useful from an educational perspective to make teaching relevant. It must be closely matched to the learning methods that exist.

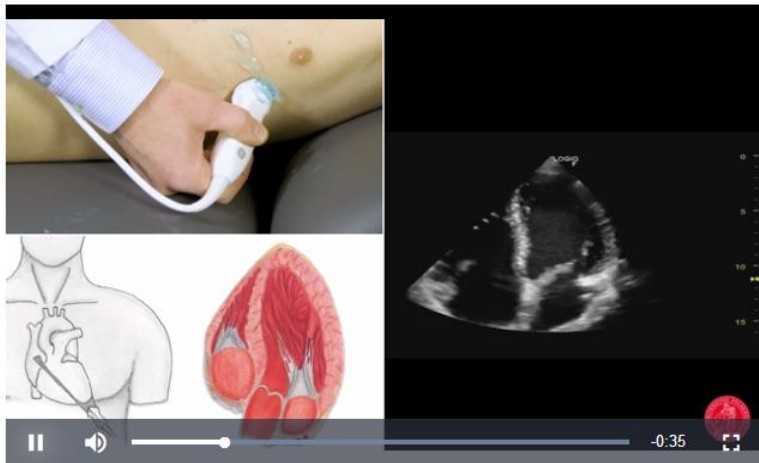

**Figure 2.** Ultrasound—an example of digitalized education.

### 4.1.3. The Alignment between Strategy and Knowledge Workers

As mentioned above, the individual teachers from each discipline are the starting point for the use of e-learning. Since the digitalization of education relies heavily on these disciplines, the activity of aligning the internal core of the organization with the strategy is very important, as discussed in the following quote:

"The question of management's ability/opportunity for "strategic management" is not specifically related to digitalization. I think Henry Mintzberg's description of universities as comparable to "pigeonholes" of diverse autonomous groupings, is relevant. This characterizes everything we do. Teaching has traditionally been "owned" by the individual teacher. Perhaps central units, in the faculties may wish for something else, but still with respect for the individual teacher's hegemony."

An example is given in this short vignette, as follows:

"In a project on implementing the BIO model (The BIO model is a way of describing the process of learning through gaining contact with the patient (beginning), gather information and summarize/plan (oppsummering in Norwegian) further treatment) we spent over a year creating a common academic understanding of clinical communication—a job that the project explicitly took on because it was not done anywhere else. This was also the case in the first edition of the movement apparatus (This is a particular project within Faculty of Medicine where e-learning resources for learning about movement apparatus (bevegelsesapparatet) were created) where the project created consensus between four subjects on how joints should be examined—a consensus that was not there before, but which should have been there. Thus, these projects can be catalysts for processes that should have been completed already".

This tight connection between discipline and changes in the discipline makes it very challenging for the managing unit to implement changes very fast.

4.1.4. Strategic Follow-Up and Maintenance

It is very difficult to obtain specific figures on how many people use e-learning within the Faculty of Medicine. The section for medical informatics does not have these numbers, and it is also very difficult to obtain specific usage figures on the solution from USIT (the central IT unit at the University of Oslo).

There are about 1000 medical students at the University of Oslo at any time, and they use e-learning to varying degrees. Usage varies very much and peaks during teaching and before exams. Teachers who have carried out an e-learning project are more likely to start a new project than novices are, and there are many disciplines with no e-learning activities at all. An explanation for this could be the amount of work required by each medical area to develop an e-learning solution. First, there is a need to acquire the needed resources (money and personnel). Then, they have to prioritize and focus. This is challenging given all the other tasks medical personnel is expected to do. These concerns are discussed as follows:

"We had a meeting a month ago, and everybody is interested in e-learning, but there are not many who use it systematically. There is a lot of work to do to establish a solution. We had to apply for money, and then Hannah [student] got money to do it ... and now we have to apply for more money for a new project we are planning."

The development is based on communication within the network, as discussed in the following statement:

"The network is used to identify potential stakeholders. It is primarily driven by enthusiasm. However, there is always a scarcity of resources, and it is difficult to identify the amount of use. Resources for e-learning can remain unaudited, there are few resources spent on follow-up. We should have had a more continuous follow-up. There will always be a cost/benefit measurement between new projects and maintenance. An example of vulnerability is that we had an ophthalmologist who was unstoppable in creating e-learning, but when she quit, all the modules and systems fell away for many years until a new enthusiast appeared. Thus, there is a major problem related to management and follow-up."

As we can see, planning and maintenance rely on the internal organization. This can be understood in light of the complex content of medicine (images, sound and video, and 3D), and the importance of high inclusion. This type of "emergent strategy" has, however, some disadvantages. The faculty has limited insight into the amount of use, and how much effort is taken to digitalize areas of each discipline. The digital infrastructure currently consists of material from 52 disciplines, but only some of the web pages are maintained regularly. The coupling between strategy and implementation is not strong enough. In addition, the initiative has some challenges related to knowledge building amongst the faculty management, since a very limited number of the personnel are familiar with the technology used. Some teachers are hoping for improvements, saying that "In practice, it will improve in a couple of years . . . we will establish learning goals for each module . . . I think this is going to be the way one chooses to acquire knowledge . . . "

There are, however, some shortcomings in the lack of continuity and follow-up, as well as the limited amount of core technology competency. The reliance on enthusiasts or knowledge brokers [41] makes the strategy vulnerable.

### 4.2. Digitalization of Sources of Law

The history of Lovdata, which is a digital infrastructure where sources of law can be looked up and interlinked through a reference system, can be traced back to the innovative activity of Jon Bing and Knut Selmer in 1970. They started an initiative called "law and data", which in 1971 was organized in a separate IT department. Lovdata, a self-financing private foundation owned by the Ministry of Justice and the Faculty of Law at the University of Oslo, was established in 1981. Using Lovdata (see Figure 3), Norway was the first country in Europe to make electronic announcements of law regulations. Students and employees at the Faculty of Law have used Lovdata in education since the late 1990s, but the classic paper collection of Norwegian acts was still the most central object for the faculty in 2017.

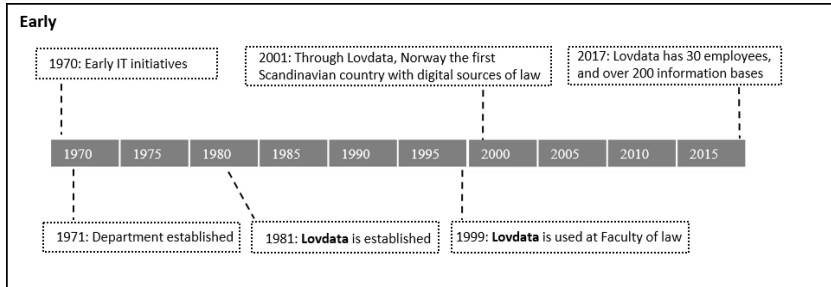
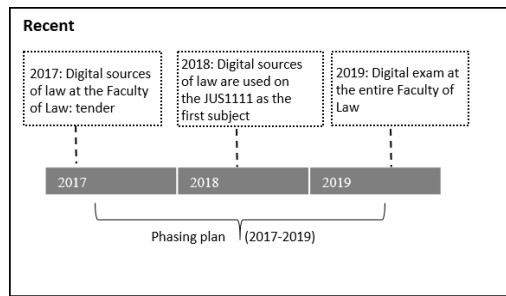

**Figure 3.** From emergent to planned strategy at the Faculty of Law.

### 4.2.1. Strategic Planning: Why Digitalize Sources of Law

The business world of lawyers requires a high level of digital expertise. For a professional lawyer, legal sources are the central tool, and "you are not an ordinary lawyer until you use what is in Database" [informant]. A central "ideal" for the Faculty of Law, thus, is the "regular lawyer who works in the business world". Students have been using Lovdata in education for some time, but until recently to a very limited degree. One of the challenges for the Faculty of Law was that students could get excellent grades even if they never entered Lovdata. The reason for this was that even though Lovdata was used in education, the exam was still performed using books, pens, and paper. Since the students are strongly motivated by the grade and, therefore, the exam, the Faculty of Law decided to change its strategy. In 2017, the Faculty of Law announced a tender, and Lovdata—which had 30 employees and over 200 information bases—needed to make changes to deal with these new requirements. Specifically, Lovdata implemented a sophisticated reference system allowing annotations and personal notes concerning the law. Moreover, Lovdata became mandatory in the exam. Earlier the students "memorized" the textbook, made notes in the

law book, and used the collection of legal judgments that were relevant to the subject. Now, they needed to also use Lovdata in their education.

### 4.2.2. Digital Resources

The primary source of law is the legal text. However, the law must be interpreted, there are ambiguities, and a hierarchy of sources of law with preparatory work (investigations and propositions), case law (supreme court, four courts of law, district courts), and legislative text (case law, administrative practice, complaints) materializes. Figure 4 is an example from "Forbrukerkjøpsloven" (Act relating to consumer purchases—Consumer Purchases Act. Section 27 Complaints) and the various references (in yellow, red, and blue) are sources that can strengthen the law practice regarding this particular paragraph in the law. The use of colors and drawings is comparable to previous paper aids but contributes by referring to related sources of law via links. This makes the use of the system dynamic and practical. The system also checks what comments and references that may be accessed on the digital exam.

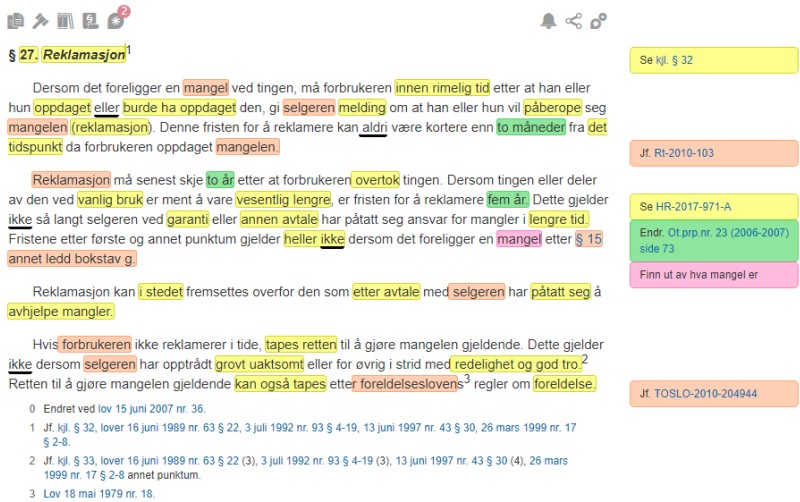

**Figure 4.** Digital sources of law in a reference system.

> "The students individualize the material through the semester, through notes, cross-teaching, and so forth. The reward is that Lovdata can be used on the exam. The practice changes the subject. Earlier the students used learning tools no one controlled, there was no clear learning strategy, and the preparation work (done through the semester) was not awarded. Now the practice of law is done more correctly, with less focus on memorizing and more rewards given to the use of juridical methods throughout the semester. The work done through the semester is rewarded. A lot of work needs to be done in advance; they cannot do everything on the exam."

According to the Dean of Education, "there is a difference between those who have been on the surface and those who seek the depth. Lovdata changes the practice".

### 4.2.3. Alignment between Strategy and Knowledge Workers

Lovdata implemented the new system in 2017. The faculty did not have many resources; however, a project group was set up to plan the introduction together with the Dean of Education. The administrative manager at the library was central in the planning of the project's training of teachers and students. The plan was to implement the system fast, but only a limited number of students were chosen to use it the first year. The course "Juss1111" was chosen as a pilot. Then, the solution was rolled out to most courses.

The strategy has met some criticism, not least from employees that protect tradition: "It is a shame if the legal faculty is in the lead to tearing down the symbol of the Norwegian

state of law—the body of law". However, the last book is due to be printed in 2018. In addition, two other universities, respectively in Bergen and Tromsø, will soon arrange a bidding process. This means that the most important universities in Norway will have digital law education. Another type of criticism comes from the student organization. They claim that they have been "overrun" and that their voice is scarcely heard. The students criticize the speed in carrying out the implementation, rather than accepting the need to digitalize law education. An informant commented, "there has also been some resistance from some of the teachers, but this depends on the generation. The younger teachers use the system right away".

### 4.2.4. Strategic Follow-Up and Maintenance

From autumn 2019, Lovdata was used in all compulsory subjects in the law study (some courses, such as criminology, as well as optional courses with other challenges that do not have an equally urgent need for legislative data, will not use it). This means that 70 courses and about 4500 students used Lovdata in the teaching and the exam at the end of 2019. Since Lovdata was required for the exam, the students used it throughout the semester. Difficulties and new requirements were taken care of by Lovdata in collaboration with the management at the Faculty of Law.

Comparing the two cases (Table 3), we see that although the strategy at the Faculty of Medicine is not without goals, the implementation and adoption rely and depend on the teachers' and the various groups' ability to implement and adopt e-learning. The faculty management uses a cultivation strategy where some funding is made available. The fast deployment and adoption rate of digital sources of law at the Faculty of Law can be explained by the structured and planned strategy that is anchored in management and based on a clear goal to improve digital competency amongst law students. The textual content makes the implementation manageable with limited resources, and a professional firm operating in the market maintains the infrastructure. In the next section, we discuss our findings.

## 5. Discussion

We see digitalization as the activity of combining physical and digital artifacts to simplify and improve processes or products [1,2]. Previous research on strategies in digitalization and infrastructures has primarily concentrated on traditional industries competing in the market [6,10,11]. According to [29], there is a strategic need for alignment between IT capabilities (in our case the digital infrastructure) and strategic goals (in our case the digitalization of higher education). To improve our understanding of how digital infrastructures [6,22] develop IS strategies [5] to solve particular challenges within higher education, we investigated two cases. We were interested in three main activities, namely the planning of the digital strategy, the alignment between the strategy and the knowledge workers, and the maintenance and governance of the digital strategy

To investigate our interests we asked the following question: how do digital strategies in higher education emerge, and how do they align with the educational context?

In the previous section, we saw that the two cases had similar intentions, namely to digitalize educational activities by using sophisticated technology. To this extent, both cases are organized according to strategies from the national authorities of education. There are, however, several differences in the cases. First, the origin of the strategic challenge may differ. It can be external [3], or internal [8]. In the law case, the requirements come from the outside, from the market, while, in the case of medicine, the internal group of teachers and students defines the requirements. This leads to a second difference. In the law case, the faculty management created a planned strategy to deal with the insufficiencies of the previous emergent strategy. This can also be understood by the differences in complexity when it comes to digital resources [30], as well as knowledge practices [31]. In law, digital resources are mainly text and references. In the case of medicine, the information content of medical images, 3D, and sound may be highly complex, and the need to include the internal

activity systems is substantial [34]. The third difference regards the activity of sustaining the digital strategy. While medicine relies on internal groups and teachers, governed by the small section for medical informatics, the Faculty of Law include Lovdata as a partner to monitor all requirements for maintenance and upgrade. These differences allow us to define two different models for digital strategy in higher education. Table 3 outlines the main differences.

The first model describes a top-down digital transformation strategy [3]. We use transformation as an outcome to depict the substantial changes in education, as well as the exam introduced by the full implementation of Lovdata. The strategy is conditioned by the commercial requirements which law firms must deal with. These requirements often emerge from the outside (for example the market), driven by the increasing need for digital competency required to become a professional lawyer. The strategic source, the digital resources, and the professional activity lay the foundations for a planned strategy.

We describe the second model as a process of bottom-up (emergent) digital innovation [2]. We use digital innovation as an outcome to emphasize that, despite the emergent and slow adaptation rate, the digital infrastructure provides a substantial amount of new digital products. Digital products are predominantly very complex and require much effort. Moreover, the strategy is conditioned by the internal organization and the requirements that emerge from this internal organization (either teachers or students).

There are also some obvious challenges with the models, especially concerning the requirements given by international and national authorities [13,16]. There is an urge to establish more effective digital relations between students and universities [42] and between the universities and the market [15]. While the first model has the advantage that it defines a strategic goal and creates a solid infrastructure to support and maintain this goal, it is less occupied with alignment between the knowledge workers and students and the strategy. The relatively fast change may cause turbulence and identity crises amongst both the employees and the students, leading to misalignments and lack of motivation.

The second model, on the other hand, relies on the internal organization to initiate innovation. The faculty aims at a loosely coupled "cultivation strategy" [8]. This may be advantageous in that change is always anchored in the organization, and that the solutions once created are compliant with educational standards. A drawback of this approach is the slow implementation pace, and the lack of planned strategy for speeding up the tempo. This is also caused by the extensive autonomy granted to the various disciplines within medicine and, hence, the difficulties of establishing a common digital foundation. The institutional reaction may be restlessness amongst the younger students and professionals, caused by outdated educational models.

To this end, we can say that while the law infrastructure is expanding to include the market (external requirements) through a more integrated and structured digital infrastructure, the e-learning infrastructure in medicine remains (very) loosely connected and within the same borders as before.

### 5.1. Theoretical Implications

Digital strategies are also increasingly invasive in higher education [12,13], and carry political, social, pedagogical, and economical requirements and aspects [15,16]. In this paper, we seek to understand the implications of centralized digital strategies on autonomous educational practices. Investigating the socio-technical implications of digital strategies [21], we observe that digital strategies sometimes lead to transformation, and other times to moderate incremental change [4]. We observe that the resulting outcome of digital strategies depends on the flexibility of the educational practices, and the elasticity of the professional objects [31]. In Table 4, we provide a comparison between two models derived from our cases. The table highlights both strategic and institutional aspects caused by different digitalization strategies.

*5.2. Practical Implications*

From a practical point of view, policymakers, politicians, educational management, and professionals need knowledge and expertise about the implications of digital strategies for organizational units. This includes understanding the implications of fast and slow digitalization. To fully understand these implications, existing educational practices are central and must be consulted. At the same time, some objects and practices are overripe for replacement.

How do digital strategies change educational practices? Moreover, who initiates the change? Is it the external or the internal stakeholders? How is this linked to social development? We see that in some cases the change is driven by commercial forces, and we describe how the organization can adapt to these requirements. At other times, the change is more conservative. This can be understood to the extent that the existing professional practices have elements in them that must be protected against fast digitalization. Central to our contribution, therefore, is the explanation for why some units are active while others are more conservative.

## 6. Conclusions

In this paper, we address the challenges and opportunities experienced by faculties when large universities aim to centralize their digitalization efforts. To address distinct implications for the faculties we investigated two classical professions, law and medicine. We aimed to identify (i) the requirements various faculties are seeking to meet through digitalization strategies, as well as the challenges their educational practices encounter in these endeavors, and (ii) the resulting solutions. We contribute by proposing two models for describing strategies in higher education. The models contain five topics concerning digital strategies. The comparison between the cases enables a more knowledgeable understanding of various professional environments choices regarding digitalization. We find that the nature of the key objects of the professional culture is central. Units, where key objects are easier to replace, may allow for a faster digitalization speed. Examples are books and paper sources that may be integrated into a common platform. When key objects are complex three-dimensional objects that require fundamental precision, digitalization may be incremental and longitudinal.

We also find that while some professional cultures adapt their educational practices to external requirements driven by commercial developments, others depend on internal changes in epistemic practices. We frame this as driven by external and internal requirements, respectively, as well as providing some pros and cons related to each strategy.

Through our socio-technical approach, we both address the strategic and institutional implications for educational practices, thus, suggesting how these two issues can be aligned.

**Funding:** This research received no external funding.

**Institutional Review Board Statement:** The project the paper is derived from was registered at, and approved by Norsk Senter For Forskningsdata (Norwegian Center for Research Data) with registration number "Referanse 823570". Approval date: 17 April 2019.

**Conflicts of Interest:** The author declares no conflict of interest.

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
