# Peer review of "Exploring the Alignment between Digital Strategies and Educational Practices in Higher Education Infrastructures"

_education, doi:10.3390/educsci12100711_

Round 1

Reviewer 1 Report

Overall, I was impressed by your paper.  The use of a literature-based conceptual framework seemed appropriate and allowed for a comparison of your two cases.  My one problem with your paper, besides the occasional subject/verb agreement issue, was your very brief conclusion. Basically, you reiterated what you intended to do in the study and did not discuss key findings and their significance.  Since there was no Discussion section in your paper, this was a major oversight.  I recommend either adding a detailed Discussion section in which you compare and contrast the two cases or, at the very least, discuss the two models the cases provide in the conclusion section. 

Author Response

Response: Thank you for the kind comments and the important observations. We have extended the conclusion, but also added theoretical and practical implications in the discussion section. We are also debating the two models in the discussion section.

Reviewer 2 Report

This research offered an interesting topic about digital transformation in education sector by taking medicine and law majors. Therefore, this research had chosen proper topics to be submitted in Education Sciences journal. I believed this manuscript can be improved much more, especially from research methods that should be detailed following the reliable methodology. Here are my review:

1. As told before, the method used in this research was not defined clearly. Suddenly, the manuscript narrated about data collection without explanation about step by step as guideline how he research worked.

2. The cas study was chosen without argumentation whether any problems occurred there or not. Why medicine and law only? Any issue about data availability?

3. Please highlight again the definition on digitalization, digital strategy, digital transformation, or any related terminologies. Check their scope to ensure relevance about definition with this research's findings.

4. Please define and state the research practical and theoretical implications.

5. Review again the research purpose? Are there any goals to compare medicine and law major? 

6. This research emphasized about alignment on digital strategies. However, how to measure or evaluation the issues on alignment? Is there any reliable methodology/technique? What are the variable that should be involved?

7. Fix the format (example: Table 1 was broken)

8. If this research adopted the longitudinal research, when was the milestone/year taken/made screenshot to create Table 1?

9. Please narrate the process to emerge Strategic Activity dan Element as shown in Table 4.

10. Suddenly, this research showed and told about first and second model. How it came from?

11. How this research actualized the triangulation process for the findings? Were they absolutely valid?

12. The latest reference was 2019. Considering digital strategies as novel topic, it should involve references that were published recently.

Thank you and have a nice revision, believe that you can improve this manuscript.

Author Response

Response Letter Education Sciences manuscript ID 1914377

Title: Exploring the Alignment between Digital Strategies and Educational practices in Higher Education Infrastructures
Journal: Education Sciences

Response Letter to Editor, Reviewers

October 2022

Dear Editors and reviewers

Thank you for your efforts in reading and commenting on this paper, and for the opportunity to submit a revised version.

Below we comment on each issue raised by the two reviewers. 

Reviewer 1:
Overall, I was impressed by your paper.  The use of a literature-based conceptual framework seemed appropriate and allowed for a comparison of your two cases.  My one problem with your paper, besides the occasional subject/verb agreement issue, was your very brief conclusion. Basically, you reiterated what you intended to do in the study and did not discuss key findings and their significance.  Since there was no Discussion section in your paper, this was a major oversight.  I recommend either adding a detailed Discussion section in which you compare and contrast the two cases or, at the very least, discuss the two models the cases provide in the conclusion section. 

Response: Thank you for the kind comments and the important observations. We have extended the conclusion, but also added theoretical and practical implications in the discussion section. We are also debating the two models in the discussion section.

Reviewer 2

This research offered an interesting topic about digital transformation in education sector by taking medicine and law majors. Therefore, this research had chosen proper topics to be submitted in Education Sciences journal. I believed this manuscript can be improved much more, especially from research methods that should be detailed following the reliable methodology.

Response: Thank you for the kind and constructive comments.

R2-1: As told before, the method used in this research was not defined clearly. Suddenly, the manuscript narrated about data collection without explanation about step by step as guideline how he research worked.

Response: We have extended the method section by adding contextual information, and the description of our analytical approach.

R2.2: The case study was chosen without argumentation whether any problems occurred there or not. Why medicine and law only? Any issue about data availability?

Response: See comment above. In the method section, we have added some comments regarding this issue.

R2.3: Please highlight again the definition on digitalization, digital strategy, digital transformation, or any related terminologies. Check their scope to ensure relevance about definition with this research's findings.

Response: Thank you for this comment. We have added definitions in the introduction section.

R2.4: Please define and state the research practical and theoretical implications.

Response: Thank you for this comment. We have added theoretical and practical implications in the discussion section.

R2.5: Review again the research purpose? Are there any goals to compare medicine and law major? 

Response: Thank you for this comment. See comment to issue 2.1. In the method section, we argue that these two classic disciplines enable a robust comparison.

R2.6: This research emphasized about alignment on digital strategies. However, how to measure or evaluation the issues on alignment? Is there any reliable methodology/technique? What are the variable that should be involved?

Response: Thank you for this comment. Our study is qualitative, and we use a theoretical framework (section 2.1) to frame our analyses. The framework provides us with key variables, but we use our data to compare the cases.

R2.7. Fix the format (example: Table 1 was broken)

Response: Fixed

R2.8. If this research adopted the longitudinal research, when was the milestone/year taken/made screenshot to create Table 1?

Response: Year added to table one (2020)

R2.9. Please narrate the process to emerge Strategic Activity dan Element as shown in Table 4.

Response: In the metod section we explain our analytical process. In the discussion section we describe the outcome, and compare the case using two generalized models. We have also added theoretical and practical implications, and extended the conclusion.

R2.10: Suddenly, this research showed and told about first and second model. How it came from?

Response: In the metod section we explain our analytical process of identifying the two models. Our models is enabled by our theoretical framework but adjusted to fit our empirical data. We discuss the models, and their emergence, in the discussion section. See also 2.1 and 2.9.

R2. 11. How this research actualized the triangulation process for the findings? Were they absolutely valid?

Response: Our method is qualitative, but we use a theoretical framework to structure our findings.
We have also debated our findings in workshops and seminars to investigate their relevance. See also 2.6

R2.12. The latest reference was 2019. Considering digital strategies as novel topic, it should involve references that were published recently.

Response: 4 new references from 2021-2022 are added

Round 2

Reviewer 2 Report

Thank you for the agile improvements. The current manuscript was enhance more than before. The suggestions were fulfilled. Finally the Abstract should capture the essential items from the manuscript. Current Abstract did not state about data collection, findings, and implications. The authors can complete them in revised abstract.

Author Response

Reviewer: Thank you for the agile improvements. The current manuscript was enhance more than before. The suggestions were fulfilled.

Response: Thank you for the kind comments

Reviewer: Finally the Abstract should capture the essential items from the manuscript. Current Abstract did not state about data collection, findings, and implications. The authors can complete them in revised abstract.

Response: New revision is submitted. The changes in the abstract is marked in the new revision.